# Corrosion Behavior and Biological Properties of ZK60/HA Composites Prepared by Laser Powder Bed Fusion

**DOI:** 10.3390/mi15091156

**Published:** 2024-09-15

**Authors:** Cijun Shuai, Cheng Chen, Zhenyu Zhao, Youwen Yang

**Affiliations:** 1School of Sino-German Robotics, Shenzhen Institute of Information Technology, Shenzhen 518172, China; 2Jiangxi Province Key Laboratory of Additive Manufacturing of Implantable Medical Device, Jiangxi University of Science and Technology, Nanchang 330013, China; 6120210206@mail.jxust.edu.cn; 3School of Mechanical and Electrical Engineering, Jiangxi University of Science and Technology, Ganzhou 341000, China

**Keywords:** magnesium alloys, hydroxyapatite (HA), corrosion resistance, ZK60, composite material, biological properties

## Abstract

Magnesium alloy ZK60 shows great promise as a medical metal material, but its corrosion resistance in the body is inadequate. Hydroxyapatite (HA), the primary inorganic component of human and animal bones, can form chemical bonds with body tissues at the interface, promoting the deposition of phosphorus products and creating a dense calcium and phosphorus layer. To enhance the properties of ZK60, HA was added to create HA/ZK60 composite materials. These composites, fabricated using the advanced technique of LPBF, demonstrated superior corrosion resistance and enhanced bone inductive capabilities compared to pristine ZK60. Notably, the incorporation of 3 wt% led to a significant reduction in bulk porosity, achieving a value of 0.8%. The E_corr_ value increased from −1.38 V to −1.32 V, while the minimum I_corr_ value recorded at 33.9 μA·cm^−2^. Nano-HA achieved the lowest volumetric porosity and optimal corrosion resistance. Additionally, these composites significantly promoted osteogenic differentiation in bone marrow stromal cells (BMSCs), as evidenced by increased alkaline phosphatase (ALP) activity and robust calcium nodule formation, highlighting their excellent biocompatibility and osteo-inductive potential. However, when increasing the HA content to 6 wt%, the bulk porosity rose significantly to 3.3%. The E_corr_ value was −1.3 V, with the I_corr_ value being approximately 50 μA·cm^−2^. This increase in porosity and weaker interfacial bonding, ultimately accelerated electrochemical corrosion. Therefore, a carefully balanced amount of HA significantly enhances the performance of the ZK60 magnesium alloy, while excessive amounts can be detrimental.

## 1. Introduction

Magnesium is gaining significant attention for bone implant fixation due to its exceptional mechanical compatibility, inherent biodegradability, and favorable biological properties, making it a promising medical metal [1]. Magnesium ions are naturally abundant in the human body and play crucial roles in numerous metabolic reactions and biological processes. The human body contains about 35 g of magnesium per 70 kg of body weight, with a daily requirement of approximately 375 milligrams. This biocompatibility enhances magnesium’s suitability for use in medical applications [2,3,4].

However, when pure magnesium is used as an implant, its naturally high hydrogen evolution rate and hemolysis rate can result in significant local hydrogen generation within the human body [5]. This disruption of the local PH balance can negatively impact the biological performance of the implant [6]. Consequently, the general use of pure magnesium in applications requiring long-term fixation, such as in the sternum, is not compatible. Alloying treatments can significantly enhance the corrosion resistance of magnesium, help mitigate the hydrogen evolution rate, and refine the microstructure of magnesium alloys by optimizing the precipitation of strengthened phases [7,8]. Despite its promising applications in medical implants, the corrosion resistance of magnesium alloys still does not fully meet stringent requirements [9]. Additionally, concerns remain regarding the potential adverse effects of these alloys, including rapid stress concentration, localized degradation rates [1], and a narrow hydrogen charge–discharge window [10], which could trigger various inflammatory and adverse reactions.

Surface modification techniques, including chemical conversion, micro-arc oxidation, coating technologies [11], and electrophoretic deposition, are effective strategies for controlling the degradation of magnesium alloys [12]. However, if the protective coating is damaged, it can lead to electric coupling between the inner and outer layers, which can accelerate the corrosion rate [13]. Another effective approach to enhance the corrosion resistance of magnesium alloys is the development of magnesium alloy composite materials [5,14,15]. These composites consist of at least two components: a matrix and a reinforcement phase. For biodegradable composites, all components must be biocompatible, biodegradable, and non-toxic. Metal-based biodegradable composites not only exhibit good mechanical properties, such as ultimate tensile strength, Young’s modulus, and yield strength, but also demonstrate excellent corrosion resistance and favorable biological properties [16].

Hydroxyapatite (HA), the primary inorganic component of human and animal bones, is known for its ability to achieve chemical bonding with body tissues at the interface. It has some solubility in the body [17], releasing harmless ions, which participate in metabolic processes. Additionally, HA stimulates bone hyperplasia and promotes the repair of defective tissue, demonstrating significant biological activity [18]. To enhance the corrosion resistance and promote the bone properties of ZK60 alloy, HA was introduced to create HA/ZK60 composite materials [19]. The addition of 15 wt% HA enabled a reduction in the corrosion rate of Mg-3Zn by approximately 60% [20]. When the HA particle content reaches 10 wt%, the ZK61-HA composite displays suitable mechanical properties, with its compressive strength and compressive yield strength being 481 MPa and 143 MPa, respectively. Furthermore, its corrosion current density (I_corr_) is approximately one-fifth that of the ZK61 Mg alloy [21]. Compared to the AZ91D matrix, the incorporation of HA particles into the AZ91D matrix significantly improved its corrosion properties, mitigated the increase in pH, and enhanced the cell viability of HBDC, MG63, and RAW 264.7 cells [22].

In the preparation of composite materials, casting methods are typically employed, encompassing extrusion casting of Mg-Sn alloy composites [23], the electromagnetic stirring casting technique for the production of graphene nanosheet-reinforced AZ91D matrix composites [24], and the combination of extrusion casting with mixing casting for the fabrication of SiCnp/Al6082 aluminum matrix composites [25], among others. However, when such composite materials are destined for use in implants, they frequently necessitate customization based on the patient’s unique condition, and the utilization of casting methods can significantly escalate production costs. Laser powder bed fusion (LPBF) is a typical additive manufacturing technology. This layer-by-layer manufacturing process allows for the easy creation of implants with controllable pore structures, enabling customization for different patients and defect sites. This study focuses on analyzing the effects of HA on the microstructure, corrosion properties, degradation properties, and osteogenic potential of ZK60, while also exploring the underlying reinforcement mechanisms.

## 2. Methodology

### 2.1. Preparation and Characterization of the HA/ZK60 Composites

To prepare the HA/ZK60 powder required for LPBF, nano hydroxyapatite (HA) was mixed with ZK60 using the mechanical ball milling method. Studies indicate that when the HA content exceeds 20%, the corrosion resistance of the local electro-couple can be significantly compromised [26]. The preparation process involved three groups, with weight percentages of 3 wt% and 6 wt% HA, respectively, which were pre-mixed using an ultrasonic vibration device. The pre-mixed powders were then placed in a ball mill tank (with a ball-to-powder ratio of 4:1) under a high-purity argon atmosphere to minimize oxidation. The powders were mixed for 1 h at a speed of 100 r/min using a high-energy ball mill (Fritsch Pulverisette-6, Idar-Oberstein, Germany). After milling, the irregular powders were separated using a sieve with an aperture of 140 μm. For clarity, the weight percentages were designated as follows: 0 wt% HA was labeled as ZK60, 3 wt% HA as ZK60/3HA, and 6 wt% HA as ZK60/6HA.

The three mixed powder groups are illustrated in Figure 1c. It is evident that the composite powders maintain a nearly spherical shape, with a smooth surface, which enhances powder fluidity. At a 3% weight percentage of HA, the nanoparticles are uniformly distributed on the surface of the spherical ZK60 powder. When the HA content reaches 6%, the nanoparticles remain evenly dispersed, though some clustering occurs on the powder surface due to van der Waals forces [27]. The morphology, size, surface characteristics, and composition of the powder significantly influence the forming quality of samples prepared by LPBF. In laser additive manufacturing, the laser reflectivity and absorptivity of the powder are critical factors that determine thermal efficiency and process stability [28]. To evaluate these properties, UV–visible near-infrared diffuse reflectance experiments [29] were conducted (UV-3600i Plus, Shimadzu, Kyoto, Japan), with results shown in Figure 1d,e. The results indicate that the laser reflectivity of the ZK60/3HA and ZK60/6HA powders is lower at a wavelength of 1064 nm compared to ZK60 powder. Consequently, the absorptivity of the composite powders is higher than that of the original ZK60 powder, as shown in Figure 1e. It is important to note that the absorption rate, being an average volume value, does not fully capture the mechanism of laser shaping.

### 2.2. Preparation and Characterization of the HA/ZK60 Product

This experiment used BLT-S210 to prepare the samples. To enhance the quality of the formed samples, a series of pilot studies were conducted to establish the optimized processing parameters: a laser power of 80 W, scanning rate of 600 mm/s, layer thickness of 30 μm, and hatching space of 60 μm. HA/ZK60 samples were manufactured using these parameters. As shown in Figure 2a, the prepared sample dimensions are 6 × 6 × 8 mm^3^. The addition of nano HA influences the absorption and reflection characteristics of the laser when interacting with the powder, which in turn affects the formation quality of the samples.

To evaluate the surface quality of ZK60 with varying HA content, the samples were first polished on sandpaper and mechanically polished with diamond polishing paste, then electrochemically polished using the ITECH DC power meter (IT6154, B + K-Precision, Yorba Linda, CA, USA). The electrolyte solution used was 20% nitric acid/alcohol solution, the working parameters were voltage (15 V), current (5 A), and electropolishing for 5~10 s. The treated samples were observed using an electron microscope (IPG’s YLR-SM Series, Figure 2b). The treated samples were subsequently observed using a gold phase microscope (GX 53, OLYMPUS, Tokyo, Japan). Subsequently, the optical micrographs of the sample surface were analyzed with Image-Pro Plus 6.0 software, from which the porosity was determined. To mitigate experimental error, at least three 50× positions were randomly selected for measurements, and the aforementioned process was repeated to analyze three layers in total.

### 2.3. Electrochemical Measurements

The electrochemical properties of the prepared samples were evaluated utilizing an electrochemical workstation to assess the impact of HA on the corrosion resistance of ZK60. Electrochemical testing employs an electrochemical workstation (PARSTAT 4000 A, Princeton Applied Research, Oak Ridge, TN, USA), in which the test sample, a platinum tablet, and AgCl are utilized as the working, auxiliary, and reference electrodes, respectively. The electrolyte comprises a homemade simulated body fluid (SBF, pH 7.4), with the specific constituents being NaCl (8.035 g/L), NaHCO_3_ (0.355 g/L), KCl (0.225 g/L), K_2_HPO_4_·3H_2_O (0.231 g/L), MgCl_2_·6H_2_O (0.311 g/L), CaCl_2_ (0.292 g/L), Na_2_SO_4_ (0.072 g/L), and (HOCH_2_)_3_CNH_2_ (6.118 g/L) [30].The processed sample was immersed in SBF for 24 h to develop a relatively stable corrosion layer on its surface, enabling the detection of a relatively stable open-circuit potential (OCP). Subsequently, electrochemical impedance spectroscopy (EIS) was conducted, spanning from high frequencies of 105 Hz to low frequencies of 10^−2^ Hz. The acquired data were analyzed and fitted using ZSimpDemo 3.30 software. Additionally, potential dynamic polarization (PDP) curve data were measured at a scan rate of 1 mV/s, with a potential range centered at the OCP ± 500 mV.

### 2.4. Immersion Experiments

In vitro immersion experiments were conducted to further investigate the degradation behavior of the samples. After grinding and polishing, the samples were soaked in 37 °C of SBF (pH 7.4) for 7 days. The ratio of the solution volume to the exposed surface of the sample was 100 mL/cm^2^. The generated hydrogen was collected into a 25 mL inverted dropper through a funnel placed above the sample. The pH change was monitored using a pH meter (FE28-Standard, Mettler-Toledo Instruments (Shanghai) Co., Ltd., Shanghai, China). To maintain a stable pH, the simulated body fluid (SBF) was replaced in a timely manner, and the volume of hydrogen generated was recorded every 12 h. The calculation formula for the corrosion rate of hydrogen evolution (*P_H_*, mm/year) is as follows:(1)$$P_{H}=2.088\times \frac{V_{H}}{t}$$
where *V_H_* (mL/cm^2^) represents the volume of hydrogen produced and *t* (day) denotes the soaking time. Following 7 days of immersion, the morphology of the corrosion products on the sample surface and cross-section was observed using scanning electron microscopy (SEM), while the composition of these products was analyzed via X-ray Diffraction (XRD). Subsequently, the corrosion products were removed with chromic acid, and the mass loss was recorded to calculate the corrosion rate (*P_W_*, mm/year):(2)$$P_{W}=2.10\times \frac{W}{At}$$

Among these parameters, *W* (mg) represents the lost mass, *A* (cm^2^) denotes the exposed surface area, and *t* (day) signifies the duration of immersion. These values are used to calculate the corrosion rate. The corrosion surface characteristic after removing the corrosion product was also observed through the SEM.

## 3. Results and Discussion

### 3.1. Microstructure

The ZK60 sample exhibited a volumetric porosity of 1.5%, with some visible fine pores. Notably, increasing the nanoscale HA content to 3 wt% resulted in a significant reduction in bulk porosity to 0.8%, along with a decrease in the number of fine pores. However, when the nano HA content increased to 6 wt%, the bulk porosity rose significantly to 3.3%. This indicates that the addition of nano HA to ZK60 can enhance its density and processing capability, but an excessive HA content may lead to increased porosity.

The observed phenomenon can be attributed to the higher laser absorption rate of the ZK60/3HA composite powder compared to that of the ZK60 alloy powder. This enhanced absorption effectively addresses the naturally high laser reflectivity of magnesium, thereby improving the forming capability of the LPBF process. Additionally, the nano-HA coating on the powder’s surface absorbs some of the laser energy and efficiently transfers it to adjacent powder particles through thermal conduction [31] This process reduces the amount of laser energy directly absorbed by the powder, minimizes magnesium vapor generation, stabilizes the melt pool, and ultimately decreases the occurrence of defects. However, when excessive amounts of nano-HA are added, the agglomeration of HA can hinder the fluidity of the molten phase within the melt pool, resulting in increased viscosity. This elevated viscosity obstructs the smooth escape of magnesium vapor and diminishes the wettability between the re-aggregated HA and the substrate, leading to a significant increase in defect formation [32]. Thus, it can be concluded that the relative density and processing capacity of LPBF-prepared parts can be enhanced only by adding 3 wt% of nano-HA to ZK60.

The phase of the HA/ZK60 composite samples was examined using XRD analysis, as depicted in Figure 2c, along with the XRD profiles of the HA nanoparticles, the ZK60 alloy, and the HA/ZK60 composite samples. The results reveal that both the ZK60 alloy and various HA/ZK60 composite samples exhibit α-Mg and Mg7Zn3 phases, while the characteristic peaks corresponding to HA are evident in the XRD pattern of the HA/ZK60 composite. Furthermore, the intensity of HA peaks in the composite samples gradually increases with the increase in the nano HA content. To investigate the specific distribution of nano-HA within the magnesium matrix, higher magnifications were observed using electron microscopy (Figure 2d). It can be observed that when the content of HA particles is low, they are uniformly dispersed within the ZK60 matrix [21]. However, the presence of HA in the composite was evident when the HA particles accounted for 6 wt%. This suggests inadequate binding at the interface between HA and the magnesium matrix. In general, an uneven distribution of enhanced particles within the matrix can negatively impact composite performance. Therefore, achieving a uniform distribution of enhanced phases in the matrix and eliminating the microscopic holes are crucial for optimizing composite material performance. This implies that selecting an appropriate HA content is essential for sample performance.

### 3.2. Electrochemical Behavior

Studies have demonstrated that HA has the potential to elicit the deposition of a calcium and phosphorus layer, thereby enhancing the protective capabilities of the corrosion layer [21]. To specifically analyze the effect of HA on the corrosion resistance of ZK60, further investigation is required. The electrochemical properties of the three samples were analyzed. Meanwhile, the formation of the corrosion layer over time was explored. Therefore, the three groups of samples were tested in EIS (electrochemical impedance spectra) to study the change in the surface corrosion layer impedance. The EIS profiles in ZK60 and HA/ZK60 composite samples at different immersion times are shown in Figure 3 (Bode plots and Nyquist plots). Since the frequency range of the EIS atlas acquired was 10^6^~1 Hz, the Nyquist atlas of all alloy samples soaked for 1, 4, and 7 days consisted of only two capacitance circuits: the capacitance rings at high and intermediate frequencies, respectively.

The high-frequency capacitance ring is caused by the charge transfer resistance on the alloy surface and the corrosion product film. The Nyquist plots reflected the presence of charge transfer resistance and Warburg resistance in the corrosion process of magnesium alloy. It is pointed out that the larger the arc radius of high-frequency capacitance, the better the corrosion resistance of the alloys will be [33]. With the soaking time, the radius of the capacitance ring becomes larger, indicating that the corrosion behavior changes during soaking. Meanwhile, the ZK60/3HA alloy has the largest capacitor ring radius, indicating that it provided the strongest protection to the alloy substrate when the HA content was 3 wt%. More specifically, the ion migration and charge transfer processes were inhibited, and the alloy had the best electrochemical corrosion resistance.

The obtained EIS results were fitted using ZsimpWin software to the equivalent circuit diagram R(Q(R(CR))) in Figure 3, and the data obtained from the fit are counted in Table 1. R_s_ in the equivalent circuit represents the solution resistance (ideally close under any conditions), R_t_ represents the charge transfer resistance, CPE_dl_ represents the electron layer constant phase element (CPE) between the corrosion product and the electrode interface (R_ct_ and CPE_dl_ are determined by the high-frequency capacitance circuit), R_f_ represents the corrosion product film’s resistance, C_f_ means the capacitance of the corrosion product film (R_f_ and C_f_ are determined by the medium frequency capacitance circuit). The CPE_dl_ is a double-layer capacitor composed of an electrolyte and a sample substrate surface that involves two important defining parameters, Y_0_ and *n*. Y_0_ indicates its non-ideal capacitance due to cracks, surface oxide film, impurity, and secondary phases. The value of *n* is the dispersion index, with values ranging from 0 to 1 and used to represent the smoothness of CPE_dl_. When *n* = 0 and 1, CPE_dl_ is the capacitance and pure resistance, respectively. According to Table 1, R_ct_ and Rf gradually increase with the soaking time. However, the ZK60/3HA alloy showed the maximum R_ct_ and R_f_ values in different immersion time periods, which showed that the surface corrosion product layer is the most stable, providing the best protection for the magnesium matrix and delaying the degradation of the magnesium matrix [34].

In the electrochemical experiments, in addition to obtaining the above EIS curves, the PDP polarization curves were measured, as shown in Figure 4a, where the shapes of the three test groups were similar. However, an obvious inflection point was shown in the anodic polarization curve of ZK60/3HA. The inflection point is called the breakdown potential (Eb), and the current density changes rapidly when it is exceeded, implying the occurrence of local corrosion [35]. The passivation phenomenon (the curve is relatively flat) can be found when the corrosion potential is lower than Eb, indicating the passivation of membrane formation, which hinders the degradation of the sample. Upon incorporating a 3% HA content, a more pronounced passivation behavior emerges, leading to enhanced protection of the corrosion product film, thereby decelerating the corrosion rate of the sample. The cathodic polarization curve that depicts the process of hydrogen evolution remains similar in shape subsequent to the addition of HA to the alloy, suggesting that an augmentation in the HA content exerts minimal influence on the cathode’s current density. However, it is imperative to observe that all these curves exhibit a rightward shift relative to the alloy cathode branch devoid of HA. Furthermore, the corrosion current density undergoes a gradual decline, suggesting that the quantity of hydrogen precipitation initially diminishes and subsequently increases with the increase in the HA content. Nevertheless, the corrosion rate is markedly slower in comparison to that of ZK60, as evident from Figure 4c,d. Due to the asymmetry of the polarization curve between the anode and cathode branches, the corrosion current density (I_corr_) and corrosion potential (E_corr_) are calculated by the cathode Tafel extrapolating method. The corresponding results are plotted in Figure 4b. As the HA content increases, the polarization curve gradually moves positively to the X axis, and the corrosion potential increases. Among these, the E_corr_ value of ZK60 was −1.38 V, while the E_corr_ of ZK60/3HA and ZK60/6HA increased to −1.32 V and −1.30 V, respectively. Notably, ZK60/3HA exhibited the smallest I_corr_, measuring 33.9 μA/cm^−2^. This indicates that the introduction of a certain amount of HA matrix can improve the corrosion resistance of ZK60.

### 3.3. In Vitro Degradation Behavior

The volume of hydrogen released from the immersion experiment over time is shown in Figure 4c. The evolution of hydrogen from ZK60 was the fastest during the immersion. In contrast, the evolution of hydrogen from ZK60 supplemented with HA increased relatively slowly. Indeed, among the three groups tested, the ZK60/3HA composite exhibited the lowest volume of hydrogen generated, suggesting it possesses the slowest degradation rate. Specifically, the corrosion rate for this composite was calculated to be 0.56 mm/year, indicating its relatively high resistance to corrosion and degradation processes. However, after the further addition of 6 wt% HA, the ZK60 showed a faster hydrogen release rate and a faster corrosion speed. Indeed, hydrogen has garnered significant attention for its therapeutic potential in treating various diseases associated with oxidative stress and inflammation. Its unique ability to act as a biological reductant and regulate homeostasis makes it a promising therapeutic agent. Hydrogen has been shown to effectively neutralize harmful reactive oxygen species (ROS) and other free radicals, thereby mitigating oxidative damage and reducing inflammation [36].

As depicted in Figure 5a, the corroded surfaces of three samples were observed using SEM after every 1, 4, and 7 days of immersion in SBF. With the extension of soaking time, all sample surfaces gradually became covered by corrosion products, which progressively increased in abundance over time. It is noteworthy that the corrosion on the ZK60 surface is the most pronounced, whereas both ZK60/3HA and ZK60/6HA exhibit relatively even corrosion surfaces. ZK60/3HA possesses the most smooth and dense corrosion surface. Spalling and cracking of corrosion products were observed in all samples due to dehydration prior to SEM observation. High-resolution SEM observation revealed that the ZK60 surface was covered with a rough and loose structure of Mg(OH)_2_. In contrast, some white particles/clusters were deposited on the surface of both ZK60/3HA and ZK60/6HA, respectively. According to previous studies, clustered spherical particles are a typical morphology of apatite. The EDS analysis shows (Figure 5c) that they are rich in Ca, P, and O, confirming that ZK60/3HA and ZK60/6HA are covered with Ca_10_(PO_4_)_6_(OH)_2_. The deposited apatite layer reduces the corrosion rate of the magnesium matrix because it has a more stable and compact structure compared to Mg(OH)_2_. The surface corrosion products were removed with a chromic acid solution consisting of chromium trioxide and silver nitrate, as shown in Figure 5b. It can be clearly seen that the surface of ZK60 is seriously corroded, with obvious local corrosion. In contrast, the corrosion of HA is more uniform, especially in the ZK60/3HA composite. However, it is worth noting that the corrosion surface of ZK60/6HA has some small corrosion pits, which may be caused by the electro-couple corrosion between the clustered HA and the magnesium matrix [37].

### 3.4. The Biological Properties of Magnesium Alloy Composites

An indirect live/dead cell staining method was used to assess the biocompatibility of the ZK60/HA composite. Staining images of bone mesenchymal stem cells (BMSCs) cultured for 1, 3, and 5 days, as shown in Figure 6a. Overall, no significant BMSCs died during culture in each group (red). With prolonged culture, the number of BMSCs increased, and the cell morphology changed from a circular shape on day 1 to a fusiform shape on day 5. The changes in the number and morphology of BMSCs reflect the suitability of the culture medium. For the same culture days, cell growth in the ZK60/3HA group was superior to that in the ZK60 group. Cell viability was determined by a CCK-8 assay, and the results are shown in Figure 6b. The cell viability of 100% extracts at 1, 3, and 5 days was 74.2%, 75.9%, and 79.1% in the ZK60/3HA group, and 61.2%, 62.8%, and 65.8% in the ZK60 group, respectively. Due to the high concentration of ions, the 100% extract was poor, but it improved in the 50% extract group. Cell viability increased to 91.2% at day 5 in the ZK60/3HA group and to 76.8% in the ZK60 group, demonstrating no cytotoxicity according to ISO criteria 10993-5:1999 [38,39].

In addition to live/dead cell staining and CCK-8 experiments, alkaline phosphatase (ALP) activity was tested in the ZK60/3HA and ZK60 (as a control) extracts. The ALP activity is commonly used to assess the degree of differentiation of BMSCs, as ALP is a marker of early osteogenic differentiation. When the cells were cultured for 7 and 14 days, they were washed three times with a phosphate-buffered saline (PBS), fixed with 4% paraformaldehyde for 15 min, and then stained with an ALP staining kit for 12 h before being observed under the microscope. The cells were harvested after 10 min of incubation with 0.05% EDTA and 0.02% EDTA (Gibco BRL, New York, NY, USA). The ALP activity was measured by the absorbance at 520 nm using a microplate reader (Beckman, Indianapolis, IN, USA). As shown in Figure 7a, after 7 and 14 days of culture, more purple areas (indicating ALP) were evident in the ZK60/3HA group, indicating that the ZK60/3HA group exhibited higher ALP activity compared to the ZK60 group. The enhancement in ALP activity was due to the introduction of HA, accelerating the release of Ca^2+^ and PO_4_^3−^, thus facilitating the mineralization process. The quantitative analysis of ALP also further verified the above conclusion, with a higher ALP content in the ZK60/3HA at both 7 and 14 days compared to the ZK60 group, as shown in Figure 7c.

In addition, the calcium nodules of the ZK60, ZK60/3HA, and ZK60/6HA groups were determined by Alizarin Red S (ARS). When the cells were cultured for 7 and 14 days, the cultured cells were fixed with 4% paraformaldehyde for 30 min. The fixed cells were cultured in a 0.5% ARS solution for 20 min in the dark. They were then observed with a light microscope. For quantification, ARS was harvested by dissolving it in 20% methanol and 10% acetic acid in water for 15 min and measured at 405 nm using a microplate reader. As shown in Figure 7b, after 7 and 14 days of culture, the number of calcium nodules (red) in the ZK60/3HA group was more than that in the ZK60 group, indicating that the addition of HA nanoparticles obviously promoted osteogenic differentiation. Subsequently, by the quantitative analysis of the ARS dyes at 7 and 14 days (Figure 7d), the amount of ARS performance increased with culture time, and higher levels of ARS performance were detected in the ZK60/3HA group compared to the ZK60 group, both at 7 and 14 days.

## 4. Conclusions

In this paper, a ZK60/HA composite sample was prepared using laser powder bed fusion (LPBF). The influence of HA on the forming quality, corrosion resistance, and biological properties of the ZK60 magnesium alloy was systematically studied. The mechanism of action was thoroughly analyzed and explained. The main conclusions are as follows: (1)The forming quality of the magnesium alloy can be evenly distributed on the surface of ZK60 spherical powder through mechanical ball grinding and adjusting the absorption and reflection of the laser. However, the introduction of excessive HA results in agglomeration on the powder surface, which is unfavorable for the formation of the magnesium alloy and instead promotes the formation of pore defects.(2)Introducing an appropriate amount of nanosized HA (3 wt.%) results in its even distribution within the matrix, providing a large number of calcium and phosphorus attachment sites and promoting the formation of an apatite protective layer. This apatite layer is dense and effectively protects the Mg matrix from further corrosion. However, excessive HA tends to cluster, and these clusters can form a local electric couple with the matrix, accelerating electrochemical corrosion.(3)ZK60/3HA not only exhibits good corrosion resistance but also provides a suitable environment for cell growth, demonstrating good biocompatibility. Furthermore, the introduction of bioactive ceramic HA enhances the release of Ca^2+^ and PO_4_^3−^, which promotes the mineralization, proliferation, and differentiation processes of bone cells, ultimately accelerating bone healing.

## Figures and Tables

**Figure 1 micromachines-15-01156-f001:**
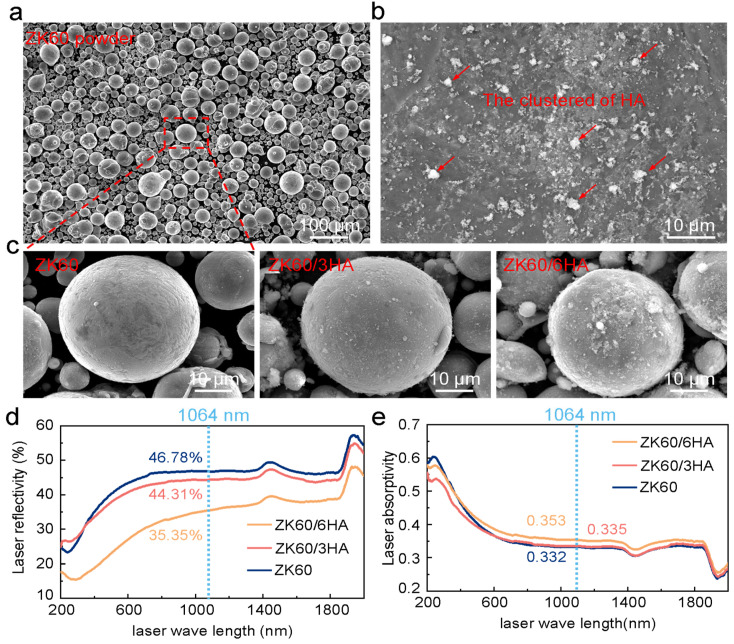
(**a**) ZK60 powders; (**b**) nano HA particles; (**c**) high-power powder morphology: ZK60 and ZK60/HA composite powder; (**d**) laser reflectivity of the composite and ZK60 powders; (**e**) laser absorptivity.

**Figure 2 micromachines-15-01156-f002:**
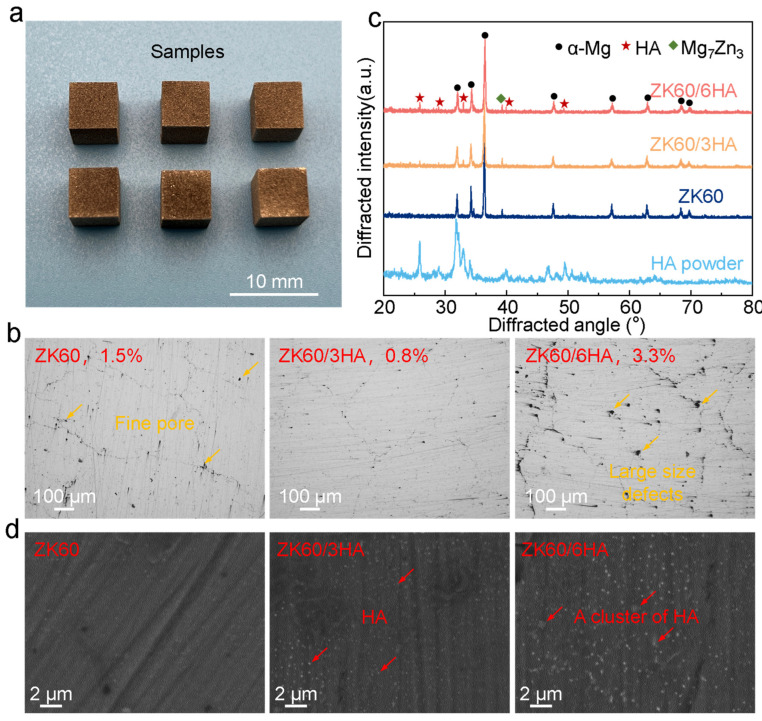
(**a**) HA/ZK60 composite samples prepared by LPBF; (**b**) surface morphology of the formed samples; (**c**) XRD map of LPBF parts and HA particles; (**d**) SEM microstructure of the formed sample.

**Figure 3 micromachines-15-01156-f003:**
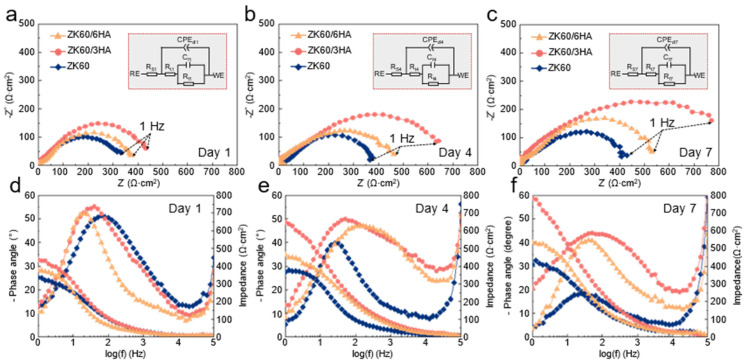
EIS profiles of different soaking times of ZK60 and HA/ZK60 composite samples prepared by LPBF: (**a**–**c**) Nyquist diagram, (**d**–**f**) Bode impedance diagram and Bode phase angle diagram, and analog circuits for different time periods.

**Figure 4 micromachines-15-01156-f004:**
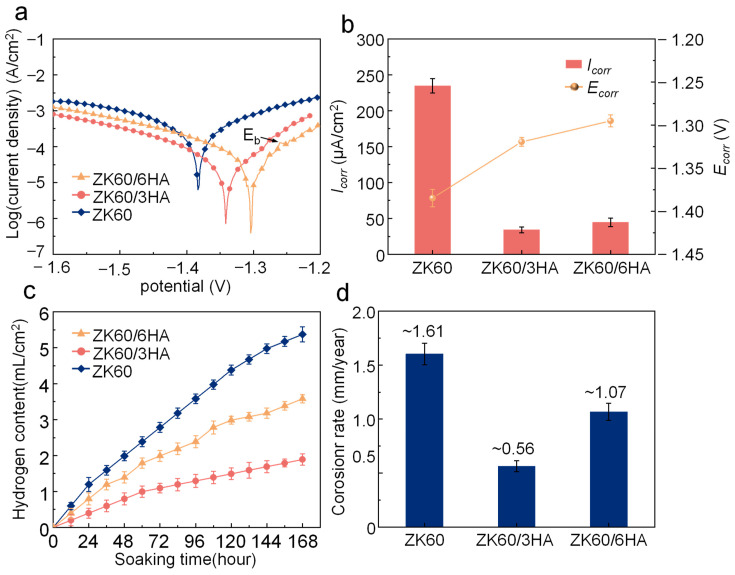
(**a**) Potential polarization curve (PDP) of ZK60/HA alloy after 7 days of immersion in SBF solution; (**b**) E_corr_ and I_corr_ obtained in the polarization curve; (**c**) change in hydrogen evolution after immersion for 7 days; (**d**) degradation rate.

**Figure 5 micromachines-15-01156-f005:**
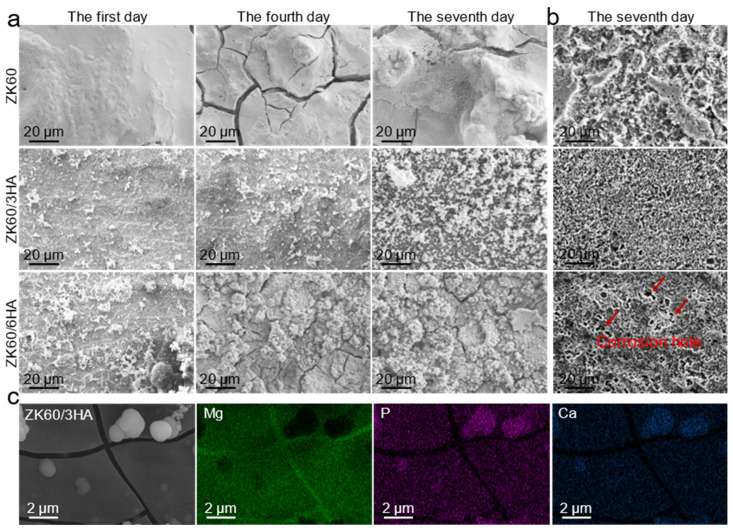
(**a**) Corrosion morphology with corrosion products after 1, 4, and 7 days in SBF; (**b**) removing the corrosion surface morphology of corrosion products; (**c**) EDS analysis of degradation products on surfaces of ZK60/3HA.

**Figure 6 micromachines-15-01156-f006:**
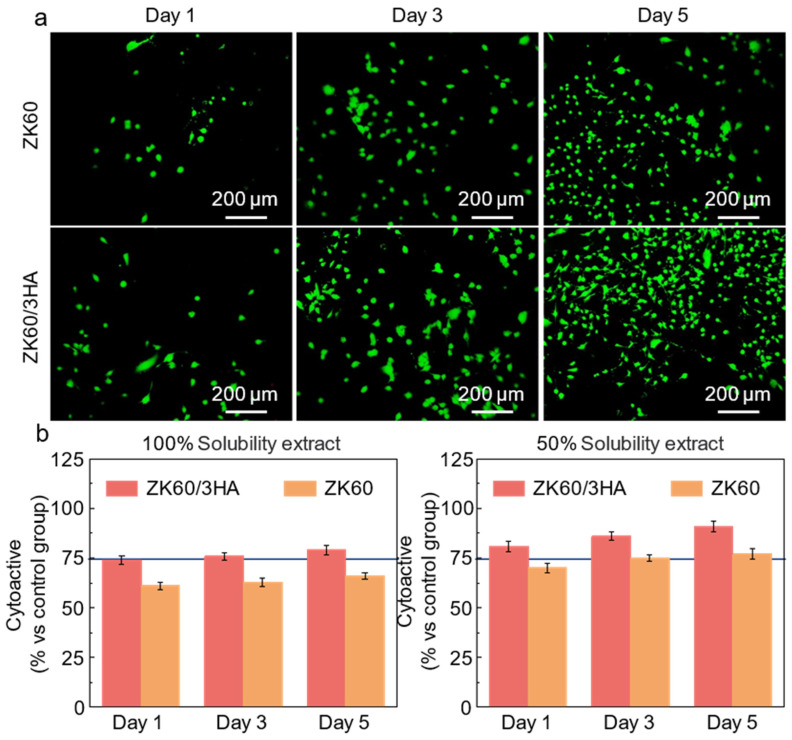
(**a**) Image of live/dead cells stained with BMSCs; (**b**) results of 100% dissolution and 50% dissolution CCK-8.

**Figure 7 micromachines-15-01156-f007:**
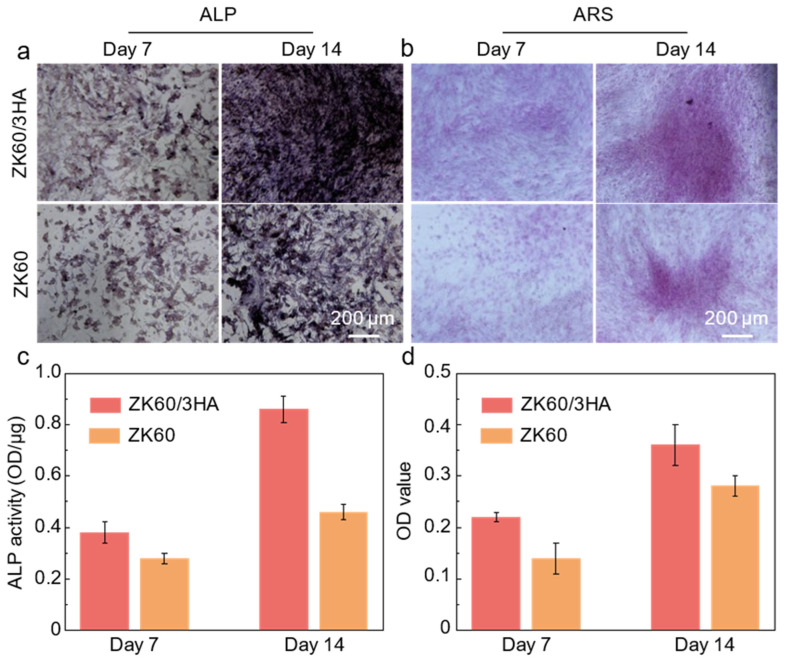
(**a**) ALP staining of BMSCs; (**b**) ARS of BMSCs; (**c**,**d**) corresponding quantitative analysis.

**Table 1 micromachines-15-01156-t001:** Fitting results of different soaking time-equivalent circuits for ZK60 and HA/ZK60 samples prepared by LPBF.

Sample	Time	*Y*_0_ × 10^−5^ Ω^−1^ cm^2^s^n^	*n*	*R_t_* (Ω·cm^2^)	*C_f_* (μF/cm^2^)	*R_f_* (Ω·cm^2^)
ZK60	One day	17.7 ± 2.3	0.5 ± 0.2	90.2 ± 3.2	15.7 ± 0.4	286.8 ± 13.2
ZK60/3HA		15.4 ± 1.6	0.4 ± 0.1	125.2 ± 8.1	9.8 ± 1.2	393.3 ± 16.2
ZK60/6HA		14.9 ± 2.1	0.5 ± 0.1	103.9 ± 4.7	12.9 ± 2.3	306.6 ± 14.7
ZK60	Four days	12.2 ± 2.1	0.6 ± 0.2	98.8 ± 6.1	13.7 ± 2.3	323.6 ± 12.5
ZK60/3HA		26.6 ± 3.2	0.4 ± 0.1	228.7 ± 1.6	8.1 ± 0.02	604 ± 26.8
ZK60/6HA		19.2 ± 2.6	0.5 ± 0.2	141.3 ± 8.7	10.8 ± 0.02	436 ± 13.5
ZK60	Seven days	11.6 ± 1.5	0.5 ± 0.2	141.8 ± 16.1	4.4 ± 1.1	355 ± 15.4
ZK60/3HA		22.7 ± 5.1	0.3 ± 0.1	303.3 ± 4.1	2.2 ± 1.3	673 ± 34.5
ZK60/6HA		17.8 ± 5.1	0.4 ± 0.1	201.4 ± 3.2	3.8 ± 1.2	489 ± 32.3

## Data Availability

The raw data supporting the conclusions of this article will be made available by the authors on request.

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
