# Peer review of "Corrosion Behavior and Biological Properties of ZK60/HA Composites Prepared by Laser Powder Bed Fusion"

_micromachines, 2024, doi:10.3390/mi15091156_

Round 1

Reviewer 1 Report

Comments and Suggestions for Authors

The purpose of this work is to investigate the ZK60/HA composite material produced by laser powder bed fusion (LPBF). The results of the study showed that the controlled addition of hydroxyapatite improved the corrosion properties of the composite. The composite also exhibited good biocompatibility. This is a current research topic and the results presented in this manuscript will be of interest to the reader. The manuscript can be accepted after minor corrections.

1. Specify the name of the equipment used for laser powder bed fusion (LPBF);

2. Specify the model of the scanning electron microscope.

Comments on the Quality of English Language

The text must be checked for errors and typos.

Author Response

The purpose of this work is to investigate the ZK60/HA composite material produced by Laser Powder Bed Fusion (LPBF). The results of the study showed that the controlled addition of hydroxyapatite improved the corrosion properties of the composite. The composite also exhibited good biocompatibility. This is a current research topic and the results presented in this manuscript will be of interest to the reader. The manuscript can be accepted after minor corrections.

1.Specify the name of the equipment used for laser powder bed fusion (LPBF);

Response:

Thank you for your comments. We have added a specification of the equipment model. All the modifications have been highlighted in yellow in the revision and as follows:

This experiment used BLT-S210 to prepare the samples. To enhance the quality of the formed samples, a series of pilot studies were conducted to establish optimized processing parameters: a laser power of 80 W, scanning rate of 600 mm/s, layer thickness of 30 μm, and hatching space of 60 μm.

  1. Specify the model of the scanning electron microscope.

Response:

Thank you for your comments. We have added a specification of the electron microscope model. All the modifications have been highlighted in yellow in the revision and as follows:

To evaluate the surface quality of ZK60 with varying HA content, polishing was per-formed, followed by observation using an electron microscope (IPG's YLR-SM Series, Fig. 2b).

Reviewer 2 Report

Comments and Suggestions for Authors

The authors conducted a study on the 'Corrosion behavior and biological properties of magnesium al-2 loy composites prepared by Laser powder bed fusion'.

At present, there are critical issues and comments that need to be addressed:

1. LPBF is highlighted in the title. However, there are no mentions of LPBF as the fabrication method in the introduction. Why LPBF? Why not other methods? Compare LPBF with other methods and justify its selection.

2. The introduction is too brief. There are insufficient discussion from previous studies on the merits of MG as the matrix material and HA as additives for a biocomposite. 

3. The lack of contents in 1. and 2. results in an unclear problem statement and motivation for this study.

4. The methodology section is too brief and does not capture the experimental procedure used in this study.

5. Some procedures described in the Results and Discussion section should really be in the Methodology section.

For detailed, section-by-section comments, please refer to the attached document.

Comments on the Quality of English Language

Some minor grammar and tense issues need to be addressed.

Author Response

Dear editors and reviewers,

Thank you very much for your letter and the comments for the manuscript [micromachines-3174025] entitled with “Corrosion behavior and biological properties of magnesium alloy composites prepared by Laser powder bed fusion”. Based on the comments and requests of reviewers, we have made a very careful modification and improvement in our revision. Our responses to the questions and comments point by point are attached at the end of the letter. We hope that our work will provide some interesting information for researchers in this field. Here, we are submitting the revision and looking forward to your further consideration for publishing the manuscript.

We appreciate your efforts and time.

Kind regards,

Youwen Yang

Jiangxi University of Science and Technology, Ganzhou 341000, China;

Email: yangyouwen@jxust.edu.cn

Reviewer#2:

At present, there are critical issues and comments that need to be addressed:

  1. LPBF is highlighted in the title. However, there are no mentions of LPBF as the fabrication method in the introduction. Why LPBF? Why not other methods? Compare LPBF with other methods and justify its selection.

Response:

Thank you for your comments. We have added LPBF as the chosen manufacturing method in the introduction section and explained why this method was chosen. All the modifications have been highlighted in yellow in the revision and as follows:

The HA/ZK60 composite was fabricated using laser powder bed fusion (LPBF), a typical additive manufacturing technology. This layer-by-layer manufacturing process allows for the easy creation of implants with controllable pore structures, enabling customization for different patients and defect sites.

  1. The introduction is too brief. There are insufficient discussion from previous studies on the merits of MG as the matrix material and HA as additives for a biocomposite.

Response:

Thank you for your comments. We added some of the characteristics of HA applied to Mg-based materials. All the modifications have been highlighted in yellow in the revision and as follows:

To enhance the corrosion resistance and promote the bone properties of ZK60 alloy, HA was introduced to create HA/ZK60 composite materials. Guo et al prepared Mg/HA composites containing 10%, 20% and 30wt.% HA, and studied its micro-organization, mechanics, corrosion resistance and biological properties in detail. In Mg/30%HA composites, HA showed significant agglomeration, which may reduce the properties of composites, but less agglomeration in Mg/20%HA composites, showing good corrosion resistance and biocompatibility. Witte et al prepared and studied the mechanical, corrosion and cytotoxicity of composites with AZ 91 as the matrix and HA as the enhanced phase. The properties of the composite materials were adjusted by changing the particle content and size.

  1. The lack of contents in 1. and 2. results in an unclear problem statement and motivation for this study.

Response:

Thank you for your comments. We add the content of 1 and 2, and clarify the research motivation of this paper.

  1. The methodology section is too brief and does not capture the experimental procedure used in this study.

Response:

Thank you for your comments. We added two parts to the methodology to make it more specific. All the modifications have been highlighted in yellow in the revision and as follows:

2.3 Electrochemical measurements

The electrochemical properties of the prepared samples were evaluated utilizing an electrochemical workstation to assess the impact of HA on the corrosion resistance of ZK60. Electrochemical testing employs an electrochemical workstation (PARSTAT 4000 A, Princeton Applied Research, USA), in which the test sample, a platinum tablet, and AgCl are utilized as the working, auxiliary, and reference electrodes, respectively. The electrolyte comprises a homemade simulated body fluid (SBF, pH 7.4), with the specific constituents being NaCl (8.035 g/L), NaHCO3 (0.355 g/L), KCl (0.225 g/L), K2HPO4·3H2O (0.231 g/L), MgCl2·6H2O (0.311 g/L), CaCl2 (0.292 g/L), Na2SO4 (0.072 g/L), and (HOCH2)3CNH2 (6.118 g/L). The processed sample was immersed in SBF for 24 hours to develop a relatively stable corrosion layer on its surface, enabling the detection of a relatively stable open-circuit potential (OCP). Subsequently, electrochemical impedance spectroscopy (EIS) was conducted, spanning from high frequencies of 105 Hz to low frequencies of 10-2 Hz. The acquired data was analyzed and fitted using ZSimpDemo 3.30 software. Additionally, potential dynamic polarization (PDP) curve data was measured at a scan rate of 1 mV/s, with a potential range centered at the OCP ± 500 mV.

2.4 Immersion experiments

In vitro immersion experiments were conducted to further investigate the degradation behavior of the samples. After grinding and polishing, the samples were soaked in 37℃ of SBF (pH7.4) for 7 days. The ratio of the solution volume to the exposed surface of the sample was 100 mL/cm2. The generated hydrogen was collected into a 25 mL inverted dropper through a funnel above the sample. The pH change was monitored using a pH meter (FE28-Standard, Mettler-Toledo Instruments (Shanghai) Co., LTD., China). To maintain a stable pH, the Simulated Body Fluid (SBF) was replaced in a timely manner, and the volume of hydrogen generated was recorded every 12 hours. Calculation formula for the corrosion rate of hydrogen evolution (PH, mm/year):

Where VH (mL/cm²) represents the volume of hydrogen produced and t (day) denotes the soaking time. Following 7 days of immersion, the morphology of the corrosion products on the sample surface and cross-section was observed using Scanning Electron Microscopy (SEM), while the composition of these products was analyzed via X-Ray Diffraction (XRD). Subsequently, the corrosion products were removed with chromic acid, and the mass loss was recorded to calculate the corrosion rate (PW, mm/year):

Among these parameters, W (mg) represents the lost mass, A (cm²) denotes the exposed surface area, and t (day) signifies the duration of immersion. These values are used to calculate the corrosion rate. The corrosion surface characteristic after removing the corrosion product was also observed through the SEM.

  1. Some procedures described in the Results and Discussion section should really be in the Methodology section.

Response:

Thank you for your comments. We have moved it to the methodology section, as shown in the fourth point.

There are also some scattered problems that we have expressed in yellow in the original article. Thank you for your comments.

Reviewer 3 Report

Comments and Suggestions for Authors

This is a very interesting paper with high relevance in the protesis 3d printing materials and its chemical reactions with the body.

The paper presents very relevant results of HA combination with the magnesium composites and also explain his best relationship  combination of of those materials.

In order to clarify some concepts, auhtor must include the 3d printing parameters used with the LPBF thecnology, to print the samples.

Author must clarify how calculated the porosity relation. Is it calculated on 2D layer? or used 3D multilayer? How many layer used for claculations?

We found same poor quality of images (red text on figure 1, A and B. Some thing in figure 2 C and D.

How calculated the corrosion rate of hydrogen evolution formula (1)?

Author Response

This is a very interesting paper with high relevance in the prothesis 3d printing materials and its Suggestions for Authors chemical reactions with the body. The paper presents very relevant results of HA combination with the magnesium composites and also explain his best relationship combination of of those materials. In order to clarify some concepts, author must include the 3d printing parameters used with the LPBF technology, to print the samples.

  1. Author must clarify how calculated the porosity relation. Is it calculated on 2D layer? or used 3D multilayer? How many layer used for calculations?

Response:

Thank you for your comments. For the detection of porosity, we are paired with multilayered polishing of the upper surface after multiregional measurements. All the modifications have been highlighted in yellow in the revision and as follows:

To evaluate the surface quality of ZK60 with varying HA content, the samples were first polished on sandpaper and mechanically polished with diamond polishing paste, then electrochemical polished using the ITECH DC power meter (IT6154, B + K-Precision, America). The electrolyte solution used is 20% nitric acid / alcohol solution, the working parameters are voltage (15 V) and current (5 A), and electropolishing for 5~10s. The treated samples were observed with an electron microscope (IPG's YLR-SM Series, Fig. 2b). The treated samples were subsequently observed using a gold phase microscope (GX 53, OLYMPUS, Japan). Subsequently, the optical micrographs of the sample surface were analyzed with Image-Pro Plus 6.0 software, from which the porosity was determined. To mitigate experimental error, at least three 50x positions were randomly selected for measurements, and the aforementioned process was repeated to analyze three layers in total.

  1. We found same poor quality of images (red text on figure 1,Aand B. Some thing in figure 2 C and D.

Response:

Thank you for your comments. We have replaced a clearer picture.

  1. How calculated the corrosion rate of hydrogen evolution formula (1)?

Response:

Thank you for your comments. As explained in the original text, the hydrogen collected in the inverted dropper was measured every 12 hours and calculated by substituting the hydrogen evolution formula for the corrosion rate, as follows:

The generated hydrogen was collected into a 25 mL inverted dropper through a funnel above the sample. The pH change was monitored using a pH meter (FE28-Standard, Mettler-Toledo Instruments (Shanghai) Co., LTD., China). To maintain a stable pH, the Simulated Body Fluid (SBF) was replaced in a timely manner, and the volume of hydrogen generated was recorded every 12 hours. Calculation formula for the corrosion rate of hydrogen evolution (PH, mm/year):

Round 2

Reviewer 2 Report

Comments and Suggestions for Authors

Thank you for addressing the comments/issues highlighted previously. While most of them are addressed adequately, the introduction section is still unsatisfactory. 

Firstly, the authors begin with describing surface modification techniques to control the degradation of Mg alloys, then continued with a more effective approach to do so via developing Mg composites. While the authors did mention about the requirements of such composites, they failed to discuss typical methods of manufacturing/fabricating such composites. The mention of LPBF at the end of the introduction section seems like an afterthought, rather than a careful consideration of the merits in selecting LPBF rather than other approaches (which are not described in the introduction section).

Secondly, the discussion on HA is still too brief. Only 2 previous studies are described in detail, considering only the effect of HA composition on the mechanical, corrosion, biocompatibility, and cryotoxicity, which are all only mentioned in general without any specificity/analysis of the detailed results. These are insufficient to highlight the merits of HA. Please analyse more related papers in detail.

Thirdly, the authors are strongly advised to include HA somewhere in the title to precisely reflect the choice of this reinforcement phase in the Mg alloy composite since they do not compare HA with other reinforcements.

Finally, some very minor issues are detected in the abstract. Please refer to the attached PDF.

Author Response

Reviewer #1:

Comments and Thank you for addressing the comments/issues highlighted previously. While most of them are Suggestions for Authors    addressed adequately, t-he introduction section is still unsatisfactory. 

1.Firstly, the authors begin with describing surface modification techniques to control the degradation of Mg alloys, then continued with a more effective approach to do so via developing Mg composites. While the authors did mention about the requirements of such composites, they failed to discuss typical methods of manufacturing/fabricating such composites. The mention of LPBF at the end of the introduction section seems like an afterthought, rather than a careful consideration of the merits in selectingLPBF rather than other approaches (which are not described in the introduction section).

Response:

Thank you for your comments. We have discussed the typical methods for preparing ZK60 composites based on your comments. All the modifications have been highlighted in yellow in the revision and as follows:

In the preparation of composite materials, casting methods are typically employed, encompassing extrusion casting of Mg-Sn alloy composites [23], the electromagnetic stir-ring casting technique for the production of graphene nanosheet-reinforced AZ91D matrix composites [24], and the combination of extrusion casting with mixing casting for the fab-rication of SiCnp/Al6082 aluminum matrix composites [25], among others. However, when such composite materials are destined for use in implants, they frequently necessi-tate customization based on the patient's unique condition, and the utilization of casting methods can significantly escalate production costs. Laser Powder Bed Fusion (LPBF), a typical additive manufacturing technology. This layer-by-layer manufacturing process allows for the easy creation of implants with controllable pore structures, enabling cus-tomization for different patients and defect sites. This study focuses on analyzing the ef-fects of HA on the microstructure, corrosion properties, degradation properties, and oste-ogenic potential of ZK60, while also exploring the underlying reinforcement mechanisms.

2.Secondly, the discussion on HA is still too brief. Only 2 previous studies are described in detail, considering only the effect of HA composition on the mechanical, corrosion,biocompatibility, and cryotoxicity, which are all only mentioned in general without any specificity/analysis of the detailed results. These are insufficient to highlight the merits of HA. Please analyse more related papers in detail.

Response:

Thank you for your comments. We have thoroughly referenced and incorporated the effects of HA components on related properties from other papers. All the modifications have been highlighted in yellow in the revision and as follows:

It has some solubility in the body[17], releasing harmless ions that participate in metabol-ic processes. Additionally, HA stimulates bone hyperplasia and promotes the repair of defective tissue, demonstrating significant biological activity[18]. To enhance the corro-sion resistance and promote the bone properties of ZK60 alloy, HA was introduced to cre-ate HA/ZK60 composite materials[19]. The addition of 15 wt% HA enabled a reduction in the corrosion rate of Mg-3Zn by approximately 60%[20]. When the HA particle content reaches 10 wt%, the ZK61-HA composite displays suitable mechanical properties, with its compressive strength and compressive yield strength being 481 MPa and 143 MPa, re-spectively. Furthermore, its corrosion current density (Icorr) is approximately one-fifth that of the ZK61 Mg alloy[21]. Compared to the AZ91D matrix, the incorporation of HA parti-cles into the AZ91D matrix significantly improved its corrosion properties, mitigated the increase in pH, and enhanced the cell viability of HBDC, MG63, and RAW 264.7 cells[22].

3.Thirdly, the authors are strongly advised to include HA somewhere in the title to precisely reflect the choice of this reinforcement phase in the Mg alloy composite since they do not compare HA with other reinforcements.

Response:

Thank you for your comments. We have included HA in the relevant titles based on your suggestion. All the modifications have been highlighted in yellow in the revision and as follows:

3.1 Microstructure of HA/ZK60 Composites

3.2. Electrochemical Behavior of HA/ZK60 Composites

3.3 In Vitro Cytotoxicity Evaluation of HA/ZK60 Composites

3.4 Biological properties of HA/ZK60 Composites

4.Finally, some very minor issues are detected in the abstract. Please refer to the attached PDF.1.Specify the name of the equipment used for laser powder bed fusion (LPBF);

Response:

Thank you for your comments. The revisions have been made according to the annotations in the PDF. All the modifications have been highlighted in yellow in the revision and as follows:

These composites, fabricated using the advanced technique of Laser Powder Bed Fusion(LPBF), demonstrated superior corrosion resistance and enhanced bone inductive capabilities compared to pristine ZK60. Nano-HA achieved the lowest volumetric porosity of 0.8% and optimal cor-rosion resistance of 0.56 mm/year.
